# Glycometabolism regulates hepatitis C virus release

Tao Yu[1,2☯], Qiankun Yang[1,2☯], Fangling Tian[1,2,3☯], Haishuang Chang[4], Zhenzheng Hu[4], Bowen Yu[4], Lin Han[1,2,3], Yifan Xing[1,2,5], Yaming Jiu[4,5], Yongning He[4,6], Jin Zhong[1,2,3]*

1 Unit of Viral Hepatitis, Institut Pasteur of Shanghai, CAS Key Laboratory of Molecular Virology and Immunology, Chinese Academy of Sciences, Shanghai, China, 2 University of Chinese Academy of Sciences, Beijing, China, 3 ShanghaiTech University, Shanghai, China, 4 Shanghai Institute of Biochemistry and Cell Biology, Center for Excellence in Molecular Cell Science, Chinese Academy of Sciences, Shanghai, 5 Cell Biology and Imaging Study of Pathogen Host Interaction Unit, Institut Pasteur of Shanghai, CAS Key Laboratory of Molecular Virology and Immunology, Chinese Academy of Sciences, Shanghai, China, 6 State Key Laboratory of Oncogenes and Related Genes, Shanghai Jiao Tong University School of Medicine, Shanghai

☯ These authors contributed equally to this work.
* jzhong@ips.ac.cn

**Data Availability Statement:** All relevant data are within the manuscript and its Supporting Information files.

**Funding:** This study was supported by the grants from Strategic Priority Research Program of the

## Abstract

HCV cell-culture system uses hepatoma-derived cell lines for efficient virus propagation. Tumor cells cultured in glucose undergo active aerobic glycolysis, but switch to oxidative phosphorylation for energy production when cultured in galactose. Here, we investigated whether modulation of glycolysis in hepatocytes affects HCV infection. We showed HCV release, but not entry, genome replication or virion assembly, is significantly blocked when cells are cultured in galactose, leading to accumulation of intracellular infectious virions within multivesicular body (MVB). Blockade of the MVB-lysosome fusion or treatment with pro-inflammatory cytokines promotes HCV release in galactose. Furthermore, we found this glycometabolic regulation of HCV release is mediated by MAPK-p38 phosphorylation. Finally, we showed HCV cell-to-cell transmission is not affected by glycometabolism, suggesting that HCV cell-to-supernatant release and cell-to-cell transmission are two mechanistically distinct pathways. In summary, we demonstrated glycometabolism regulates the efficiency and route of HCV release. We proposed HCV may exploit the metabolic state in hepatocytes to favor its spread through the cell-to-cell transmission *in vivo* to evade immune response.

## Author summary

Hepatitis C virus (HCV) is a positive-stranded RNA virus that causes acute and chronic hepatitis and hepatocellular carcinoma. HCV infectious cycle comprises viral entry, uncoating, translation and replication of viral RNA, assembly into new virions and release. Establishment of HCV cell culture system (HCVcc) has yielded many insights into complete HCV infectious cycle in Huh7 cell and Huh7-derived human hepatoma cell lines. However, because hepatoma-derived cell lines and hepatocytes vary in metabolism,

Chinese Academy of Sciences (XDB29010205) and the National Natural Science Foundation of China (32070944) to JZ. The funders had no role in study design, data collection and analysis, decision to publish, or preparation of the manuscript.

HCV infectious cycle in tumor cell lines and the patient's liver may also be different. Therefore, we explored the alterations of HCV infectious cycle by forcing the tumor cell lines to switch their glycometabolic pathways. We found that HCV release can be blocked by culturing cells in galactose-containing medium, leading to accumulation of intracellular infectious virions within MVB. Moreover, we provided new evidence to suggest that HCV cell-to-cell transmission may be mechanistically distinct from cell-to-supernatant release. Finally, we proposed a new concept that HCV release from hepatocytes into circulation may be naturally inefficient due to the metabolic state in liver that may favor more HCV cell-to-cell transmission. This strategy would allow HCV to effectively evade neutralizing antibodies to establish persistent infection.

## Introduction

Hepatitis C virus (HCV) is an enveloped positive-stranded RNA virus that belongs to the *Flaviviridae* family. The 9.6-kb viral genome contains a single open reading frame (ORF) which encodes a polyprotein of approximately 3,000 amino acids. The open reading frame is flanked at the 5' and 3' ends by highly conserved non-coding regions (NCRs) which play important roles in HCV protein translation and genome replication. Translation of the HCV open reading frame yields a polyprotein precursor that is posttranslationally processed into structural (Core, E1, and E2) and nonstructural (p7, NS2, NS3, NS4A, NS4B, NS5A, and NS5B) proteins by host and viral proteases [1]. HCV infection tends to be persistent, which results in liver cirrhosis and hepatocellular carcinoma [2].

HCV infectious cycle comprises viral entry, uncoating, translation and replication of viral RNA, assembly into new virions and release. HCV enters hepatocytes through receptor-mediated endocytosis [3]. The viral genome is multiplied in replication complexes located on the cytoplasmic face of the endoplasmic reticulum (ER) [1]. The encapsidation of nascent viral RNA genome by the viral core protein takes place on the surface of lipid droplets (LDs) [4]. The assembled viral ribonucleoprotein (RNP) complex relocates to the ER membrane and buds into the ER lumen during which the enveloped virions are formed. The association of viral particles with lipoproteins contributes to HCV infectivity and release [5]. The mature HCV virions released from hepatocytes are associated with very-low-density lipoprotein (VLDL) to form lipoviroparticles [6]. It has been reported that HCV is actively secreted by infected cells through a Golgi-dependent mechanism while bound to VLDL [7]. Blockage of VLDL assembly or secretion by siRNA targeting apolipoprotein B (apoB) or apoE reduces HCV production, suggesting the association between HCV and VLDL is important for virus infectious cycle [7,8]. However, the role of VLDL in HCV release remains controversial. Interfering with or dominant negative manipulation of Rab GTPases that are critical for trans-Golgi network-endosome trafficking impairs HCV release, but does not have a concomitant effect on secretion of triglycerides, apoB or apoE [9]. Knockdown or pharmacological inhibition of clathrin or clathrin adaptor AP-1 impairs HCV release without altering intracellular HCV levels or apolipoprotein B (apoB) and apoE exocytosis [10]. These reports suggest that VLDL participates in HCV assembly/release, but these two pathways do not completely overlap. Recent evidence indicates hepatic exosomes can transmit productive HCV infection *in vitro* and are partially resistant to antibody neutralization [11], suggesting a potential role of multivesicular body (MVB), the intracellular reservoir of exosomes, in HCV release and transmission [12,13]. However, the details of HCV release process still need to be further explored.

Establishment of HCV cell culture system (HCVcc) has yielded many insights into complete HCV infectious cycle in Huh7 cell and Huh7-derived human hepatoma cell lines [14–16]. Unlike normal cells, however, tumor cells undergo unusually high rate of glycolysis instead of oxidative phosphorylation even in the presence of sufficient oxygen. This aerobic glycolysis in tumor cells is referred to as "Warburg effect" [17]. Galactose enters glycolysis through the Leloir pathway and generates glucose-6-phosphate (G-6-P) less efficiently than glucose, leading to the reduced glycolysis [18]. It has been reported that tumor cells grown in galactose-containing medium are forced to change their energy metabolism from aerobic glycolysis to oxidative phosphorylation [18–20]. This glycometabolic change has a significant impact on many cellular events. For example, tumor cells grown in galactose-containing medium are susceptible to nitric oxide attack and mitochondrial toxicants [19,21]. The ability to produce IFN-γ is markedly compromised when activated T cells are blocked from engaging glycolysis upon galactose medium culturing [22].

In this study we sought to assess the effect of glycometabolism on HCV infection by culturing the hepatoma-derived cells in the glucose- or galactose-containing medium. We found that glycometabolism in the host cells regulates efficiency and routes of HCV release. We proposed that HCV may exploit the metabolic state of hepatocytes *in vivo* to minimize its free-virus release efficiency while maintain the cell-to-cell transmission to evade host immune response.

## Results

### HCV release is significantly reduced when cells are cultured in galactose medium

Since tumor cells cultured in galactose-containing medium are forced to use oxidative phosphorylation for the energy production and produce much less amount of lactate compared with the cells cultured in glucose-containing medium (S1A Fig) [19], we firstly sought to determine whether culture in galactose-containing medium affects HCV infection in Huh7 hepatoma cell line. Huh7 cells cultured either in glucose or galactose were inoculated with HCVcc (JFH1 strain) [16] at a low multiplicity of infection (MOI) of 0.01. Both HCV extracellular infectivity and intracellular RNA were significantly lower in the cells cultured in galactose (Fig 1A and 1B). To investigate which step(s) of HCV infectious cycle was affected in galactose-containing medium, we analyzed the viral entry and genome replication by using HCV-pseudotyped particles (HCVpp) that mimic HCV envelope glycoprotein-mediated entry process [23,24] and HCV subgenomic replicon (JFH1-SGR) that recapitulates viral replication and translation processes [15,25,26] respectively. The HCVpp infection (Fig 1C), NS3 protein (Fig 1D) and HCV RNA levels (S1B Fig) in the replicon cells were comparable between the two groups, suggesting that HCV entry and genome replication are not affected by the galactose medium. Next, we examined the impact of the galactose medium in a high MOI (MOI = 2) infection experiment in which majority of the cells would become infected after the initial inoculation to exclude an accumulative effect from the multi-round infection. We found galactose did not significantly affect HCV NS3 and Core protein expression (Fig 1E), but significantly reduced extracellular infectious virion production (Fig 1F), extracellular HCV RNA level (S1C Fig) as well as specific infectivity (S1D Fig). To our surprise, intracellular infectivity titers were significantly higher in the galactose-cultured cells (Fig 1G), suggesting that HCV release might be impaired in galactose medium. Consistently, 2-Deoxy-D-glucose (2DG), a glucose analog that inhibits glycolysis [27], significantly decreased the extracellular HCV titer (Fig 1I) and increased intracellular HCV titer (Fig 1J) in glucose medium, while it had little effect on HCV replication (Fig 1H).

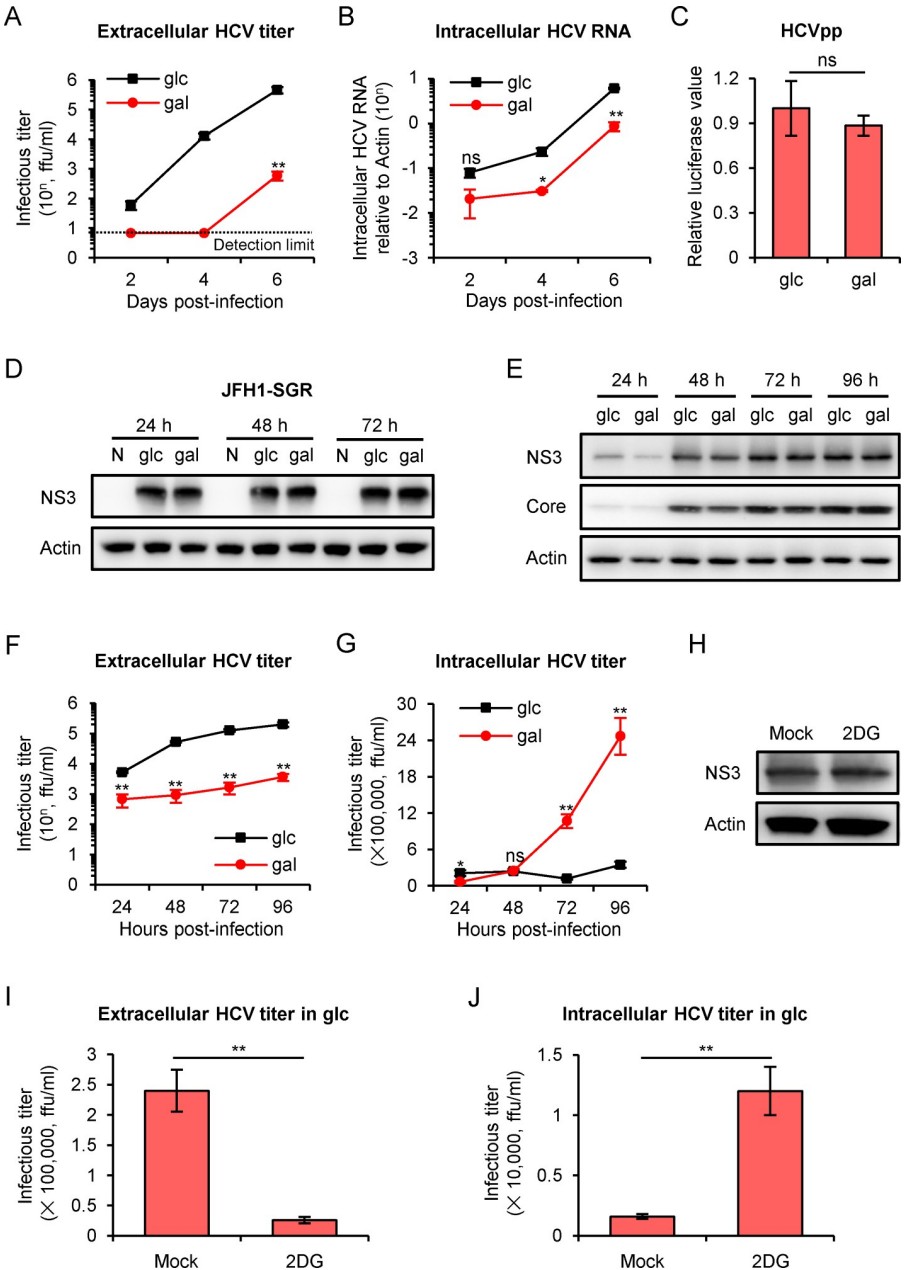

**Fig 1. Galactose medium culturing decreases HCV virion production but has no effect on virus replication and entry. (A-B)** Kinetics of extracellular HCV titers and intracellular HCV RNA levels following a low MOI infection. Huh7 cells that had been pre-cultured in glucose or galactose medium for 12 hours were infected with HCV at MOI of 0.01. The titers and RNA levels were measured at day 2, 4 and 6 post-infection. **(C)** HCV pseudotyped particles (HCVpp) assay. Huh7 cells that had been pre-cultured in glucose or galactose medium for 12 hours were infected with pseudotyped viruses bearing JFH1 envelope proteins glycoproteins. HCVpp infection was measured by the luciferase assay at day 3 post-infection. **(D)** Western blot of HCV NS3 proteins in naïve Huh7 cells (N) and Huh7 cells harboring JFH1 subgenomic replicon (SGR) cultured in glucose or galactose medium for 24, 48 and 72 hours. **(E-G)** Huh7 cells that had been pre-cultured in glucose or galactose medium for 12 hours were infected with HCV at MOI of 2. Viral NS3 and Core protein levels (**E**), extracellular titer (**F**) and intracellular titer (**G**) were examined at 24-, 48-, 72- and 96-hour post-infection. **(H-J)** Huh7 cells that had been cultured in glucose medium supplemented with 4 mM 2DG for 12 hours were infected with HCV at MOI of 2. Viral NS3 protein (**H**), extracellular titer (**I**) and intracellular titer (**J**) were examined at 72-hour post-infection. Data were presented as the mean ± standard deviation (error bars) of triplicate. glc: glucose medium; gal: galactose medium.

Since Huh7 cells grew faster in glucose than in galactose (S1E Fig), reduction of HCV release in galactose medium may possibly result from an indirect and accumulative effect of the cell proliferation during a long course of experiment. To rule out this possibility, we designed a medium-switch experiment. Galactose-cultured Huh7 cells were infected with HCVcc at an MOI of 2 for 48 hours, and then the culture medium was switched to glucose or left in galactose (S2A Fig). The cell proliferation remained at comparable levels between glucose and galactose within the first 24 hours after the medium switch (S2B Fig), while HCV release increased significantly as early as 12 hours after the switch (S2C Fig). Consistently, while the intracellular infectivity titers in the galactose-cultured cells continued to increase, they dropped at 24-hour after culture medium was changed to glucose (S2D Fig). HCV Core protein expression remained unchanged after the medium switch (S2E Fig), again confirming that the viral genome replication and protein translation were not affected. Conversely, medium switching from glucose to galactose (S3A Fig) led to a rapid suppression of HCV release as early as 12 hours (S3B Fig) and accumulation of infectious viral particles within the cells (S3C Fig). These results suggested that HCV release process is rapidly regulated by the carbohydrate source in the culture medium. Interestingly, culturing cells in galactose-containing medium had no effect on the production of ZIKA virus (ZIKV) and Dengue virus (DENV) that both belong to the same family of *Flaviviradae* as HCV (S4 Fig).

HCV virion maturation and release overlap with the VLDL secretory pathway [28,29]. Since the sterol regulatory element-binding protein 1 (SREBP1), the key master regulator of host lipid metabolism, is regulated by glucose at the transcriptional level [30], we determined whether VLDL secretion was affected by the carbohydrate source in the culture medium. We found that glucose- and galactose-cultured cells secreted similar levels of VLDL (S5A Fig) or apoE (S5B Fig), an important protein component of VLDL and essential for HCV infectivity. In addition, the cells cultured in the two media had similar levels of SREBP1/2 mRNA (S5C Fig), active form of SREBP2 protein (S5D Fig), mRNA of SREBP1/2 target genes (SCD1, ELOV6 and ACACA) (S5E Fig) and intracellular cholesterol (S5F Fig), suggesting that inhibition of HCV release in galactose is not due to a general blockade of host VLDL secretory pathway.

## Intracellular HCV infectious particles are trapped within MVBs

To investigate the subcellular localization of accumulated HCV virions within the galactose-cultured Huh7 cells, we performed transmission electron microscopy. There were significantly more multivesicular bodies (MVBs) in the galactose-cultured cells than in the glucose-cultured cells regardless of HCV infection (Figs 2A and S6A). The expression of CD63, an important MVB marker [31], also increased in the galactose-cultured cells analyzed by Western blotting (Fig 2B), immunofluorescence (Fig 2C) and flow cytometry (S6B Fig). Confocal microscopy showed that CD63 was dispersed in the cytoplasm in the glucose-cultured cells, but concentrated near the nucleus in the galactose-cultured cells (S6C and S6D Fig). Furthermore, anti-E2 immuno-gold labeled HCV-like particles can be found in MVBs of galactose-cultured cells (Fig 2D).

MVB contains multiple intra-luminal vesicles (ILVs) (Fig 2A insets and S6A Fig), which are formed by the inward budding of late endosomal membrane. The fusion of MVB with the plasma membrane leads to the secretion of the ILVs into the extracellular environment as exosomes [32]. We therefore speculated exosome secretion in the galactose-cultured cells would also decrease. To test this hypothesis, we isolated exosome from culture supernatants, and determined the levels of exosomal marker protein CD63 by Western Blotting. As expected,

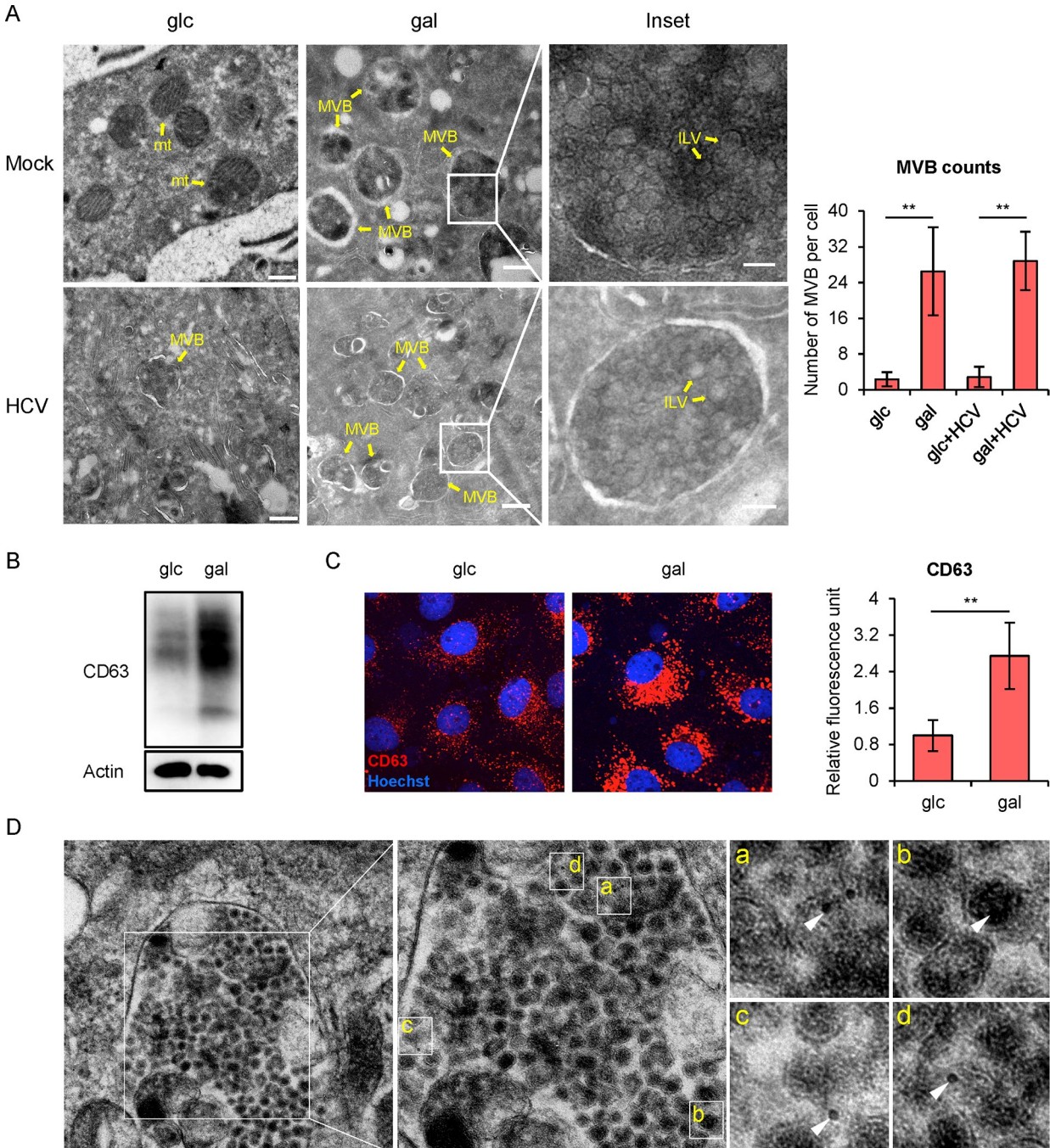

**Fig 2. HCV virions are accumulated within MVBs of the galactose medium-cultured cells.** **(A)** Electron microscopic images of the glucose- or galactose-cultured Huh7 cells that had been infected by HCVcc at MOI of 2 or mock infected for 2 days. Multivesicular bodies (MVBs) and mitochondria (mt) in cytoplasm were labeled by arrows (scale bars 500 nm). Intraluminal vesicles (ILVs) in MVBs were shown in insets (scale bars 100 nm). The average MVB number per cell was counted and shown on the right. The error bars represented standard deviation from the measurements in at least 10 cells. **(B)** Western blot of CD63 protein in Huh7 cells cultured in glucose or galactose medium for 48 hours. **(C)** Confocal immunofluorescent analysis of CD63 (red) in Huh7 cells that had been cultured in glucose or galactose medium for 48 hours. Nuclei were stained with Hoechst (blue). The relative integrated optical density (IOD) of CD63 signals were quantified by Image-Pro Plus 6.0 and shown on the right. The error bars represented standard deviation from the measurements in at least 30 cells. **(D)** Electron microscopy of the immuno-gold (arrow) labeled MVB by anti-E2 antibodies (left, bar 200 nm; middle, bar 100 nm); The immuno-gold labeled HCV-like particles in MVB are shown on the right (bar 30 nm). glc: glucose medium; gal: galactose medium.

galactose-cultured cells secreted much less CD63 than glucose-cultured cells (S6E Fig). Consistently, CD63 secretion was also dramatically decreased in 2DG-treated cells (S6F Fig).

To confirm the infectious HCV virions are indeed trapped in MVBs, we performed a density fractionation assay. HCV-infected Huh7 cells cultured in the glucose- or galactose-medium were lysed, and the nucleus-depleted lysates were subjected to the buoyant density ultracentrifugation (Fig 3A). There were two peaks of CD63 –fraction #4 and #8. While in the glucose-cultured cells more CD63 proteins were distributed in fraction #4 where the lysosome marker LAMP1 was mainly located, CD63 distribution in the galactose-cultured cells was shifted to fraction #8 where infectious HCV particles were also accumulated (Fig 3B–3D). Electron microscopy analysis of the contents in fraction #4 and #8 showed that they contained lysosome-like vesicles and MVB-like vesicles respectively (Fig 3E). Taken together, these results demonstrated that intracellular infectious HCV particles are trapped in MVBs of the galactose-cultured cells.

## Inhibition of the MVB-lysosome fusion promotes HCV release in the galactose medium-cultured cells

Previous studies demonstrated that the fusion of MVB with lysosome would promote degradation of MVB contents and reduce exosome secretion [32]. Phosphatidylinositol-3,5-bisphosphate [PI(3,5)P$_2$] is a low-abundance phosphoinositide that plays important roles in mediating MVB targeting to lysosome [33,34]. PI(3,5)P$_2$ is synthesized from phosphorylation of PI3P catalyzed by a phosphatidylinositol 3-phosphate 5-kinase PIKfyve [33], which negatively regulates exocytosis in neurosecretory cells and pancreatic acinar cells [35,36]. Inhibition of PIKfyve increases exosome secretion through reducing MVB-lysosome fusion [37]. To determine the impact of MVB-lysosome fusion on HCV release, we treated HCV-infected Huh7 cells with apilimod, a PIKfyve inhibitor [38]. The apilimod treatment increased HCV release in galactose-cultured Huh7 cells in a dose-dependent manner, while had no significant effect on HCV release in glucose-cultured cells (Fig 4A and 4B). We noted that apilimod, even at a low concentration, slightly inhibited HCV replication in both media (Fig 4A and 4B). To confirm the results of the chemical inhibitor treatment, we performed a PIKfyve knockdown experiment. Huh7 cells cultured in glucose or galactose medium were infected with HCVcc at an MOI of 2 for 24 hours, and then transduced with lentiviruses expressing control shRNA or PIKfyve-specific shRNA for another 3 days. As shown in Fig 4C and 4D, while knockdown of PIKfyve inhibited HCV replication in both media, it promoted HCV release in the galactose-cultured cells but not in the glucose-cultured cells. Together, these data suggested that the fusion of MVB with lysosome contributes to blockade of HCV release in galactose medium.

## Pro-inflammatory cytokines enhance HCV release in the galactose medium-cultured cells

Previous studies showed intracerebral injection of interleukin 1 beta (IL-1β) induces the secretion of extracellular vesicles from astrocytes into peripheral circulation and promotes the transmigration of leukocytes through modulation of the peripheral acute cytokine response [39]. It has also been shown that tumor necrosis factor α (TNF-α) enhances the release of exosomes by tumor cells [40]. Therefore, next we tested whether the pro-inflammatory cytokines IL-1β and TNF-α affect HCV release in glucose- or galactose-containing medium. The antiviral cytokine interferon-α (IFN-α) was also included as a control. We found that although IL-1β and TNF-α had a minor or no significant effect on intracellular HCV RNA levels in either medium, they enhanced HCV release specifically in the galactose-cultured cells but not in the glucose-cultured cells (Fig 5A–5D). As expected, antiviral IFN-α significantly inhibited HCV

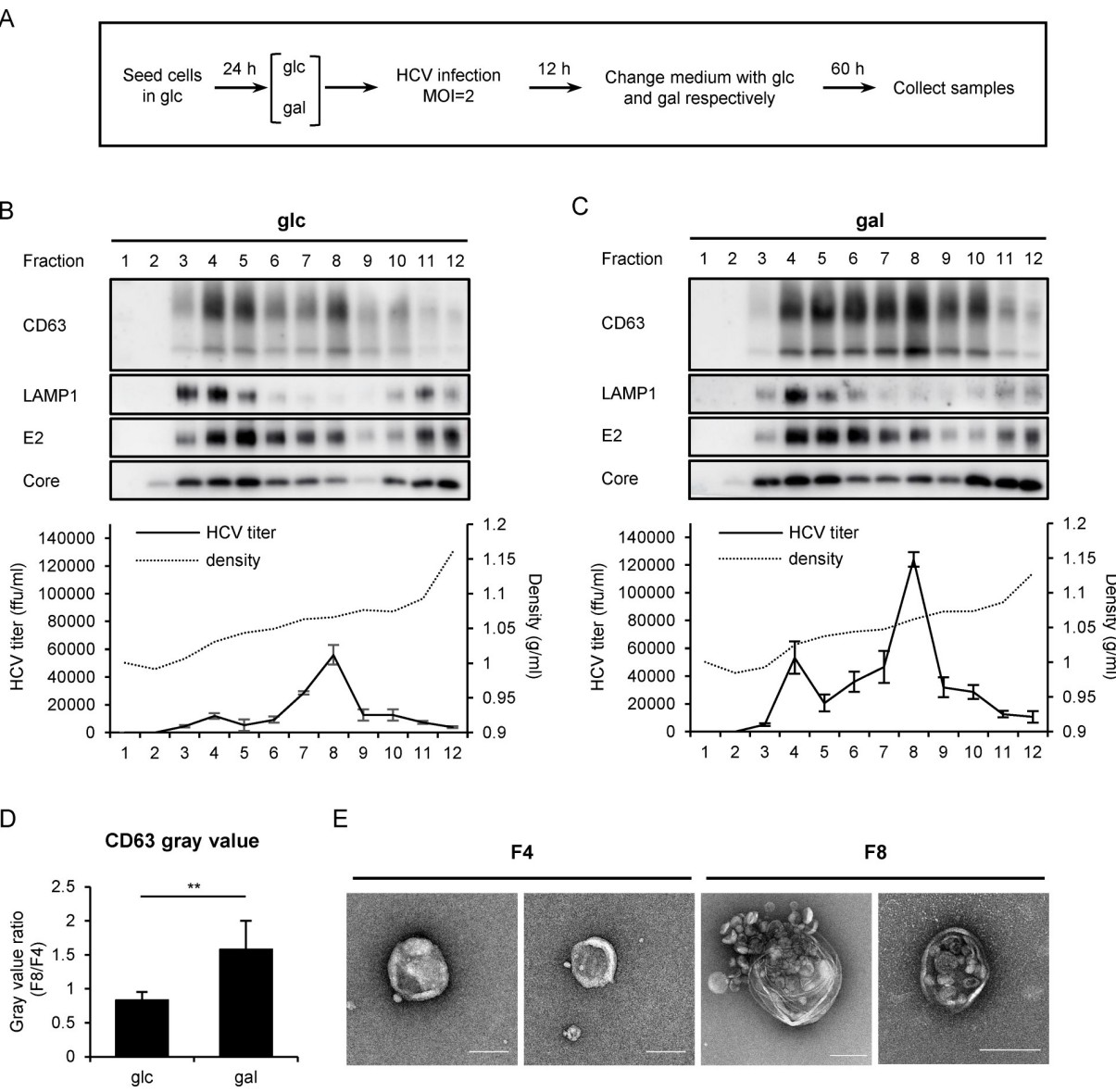

**Fig 3. Intracellular HCV virions were co-fractionated with MVBs. (A)** Schematic of the protocol to prepare the samples for the iodixanol density ultracentrifugation. **(B-C)** Western blot of CD63, LAMP1, HCV E2 and Core in each fraction collected from ultracentrifugation of HCV-infected Huh7 cells cultured in glucose **(B)** or galactose **(C)**. HCV titer and iodixanol density in each fraction were shown below the blots. Error bars were derived from three measurements. **(D)** The quantification of the CD63 protein levels in fractions 4 and 8 from three independent ultracentrifugation experiments of **(B)** and **(C)**. The CD63 levels were quantified by Image J and expressed by a ratio of fraction 8 versus fraction 4 (F8/F4). **(E)** Transmission electron microscopy of the contents in fraction #4 and #8. Two representative images were shown in each fraction (bar 100 nm). glc: glucose medium; gal: galactose medium.

extracellular titers and intracellular RNA levels in both media (Fig 5E and 5F). Next, we performed a time course experiment in which 10 ng/ml of IL-1β promoted HCV release in the galactose-cultured cells as early as 24 hours post-treatment (Fig 5G and 5H). Together, these data suggested a potential relationship among glycometabolism, pro-inflammatory signaling and HCV release. Consistently, IL-1β treatment significantly promoted exosomal CD63 release in the galactose-cultured cells (S7 Fig).

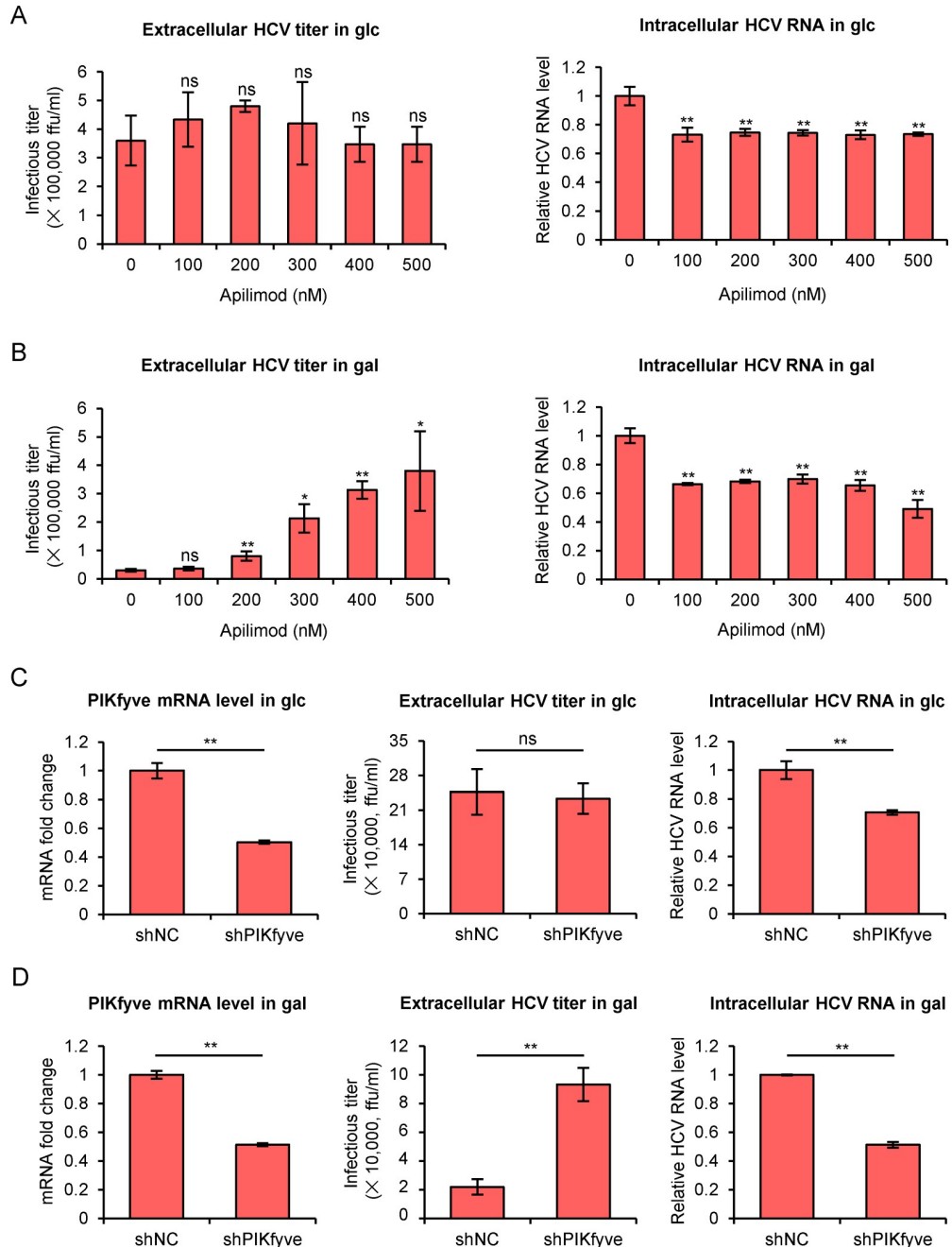

**Fig 4. Inhibition of the MVB-lysosome fusion promotes HCV release in the galactose medium-cultured cells.**
**(A-B)** Huh7 cells that had been cultured in glucose (**A**) or galactose (**B**) medium for 12 hours were infected with HCV at MOI of 2 for 1 day and then treated with different concentrations of apilimod (0–500 nM). Extracellular HCV titers and intracellular HCV RNA were determined at 24-hour after the apilimod treatment. **(C-D)** Huh7 cells that had been cultured in glucose (**C**) or galactose (**D**) for 24 hours were infected with HCV at MOI of 2 for 1 day, and then transduced with lentiviruses expressing control shRNA (shNC) or shRNA targeting PIKfyve (shPIKfyve). PIKfyve mRNA level, extracellular HCV titer, and intracellular HCV RNA level were determined 3 days after lentivirus infection. Data were presented as the mean ± standard deviation (error bars) of triplicates. glc: glucose medium; gal: galactose medium.

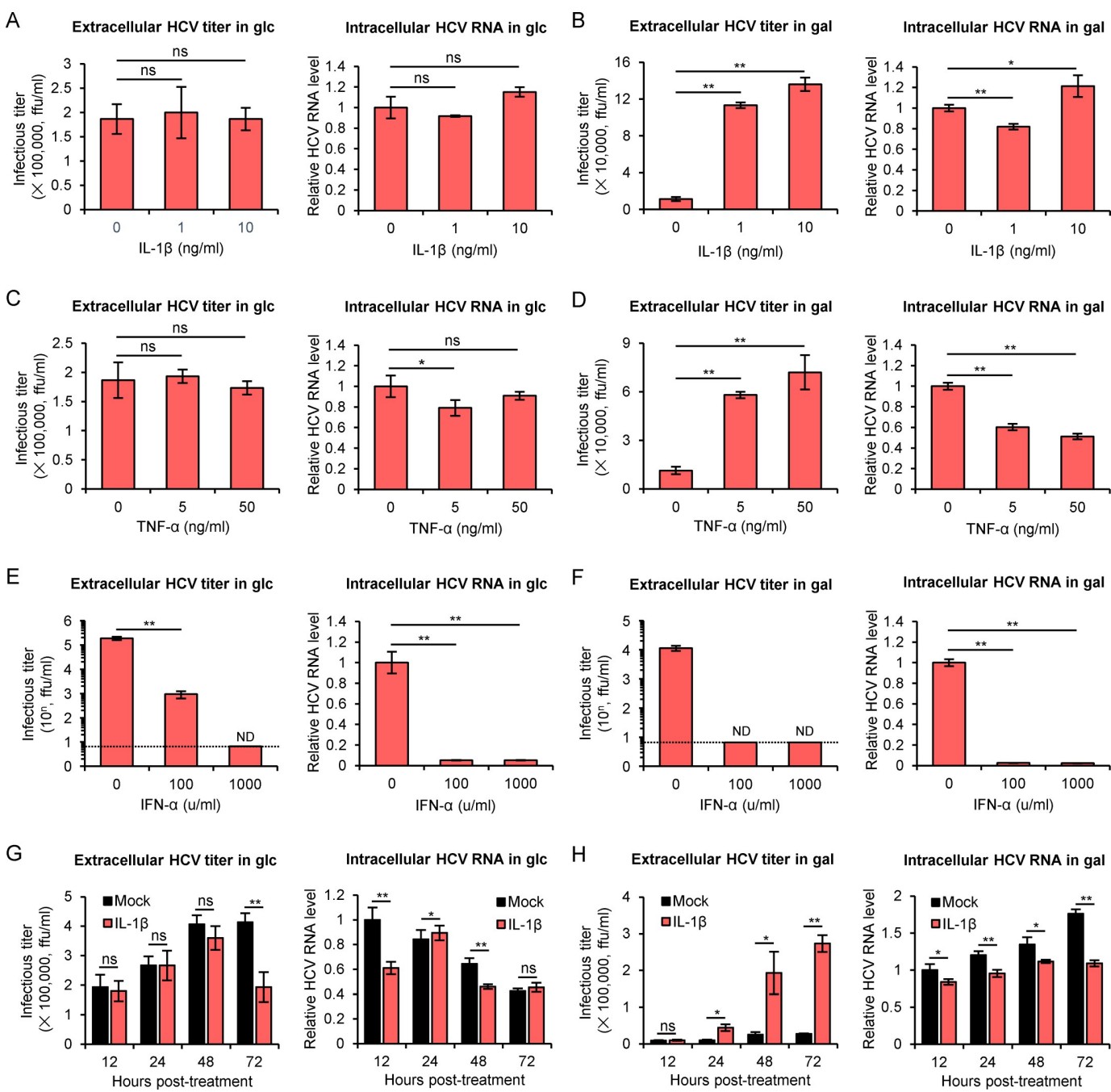

**Fig 5. Pro-inflammatory cytokines enhance HCV release in the galactose medium-cultured cells. (A-F)** Huh7 cells that had been cultured in glucose (**A, C, E**) or galactose (**B, D, F**) for 12 hours were infected with HCV at MOI of 2 for 1 day, and then treated with IL-1β (**A, B**), TNF-α (**C, D**) or IFN-α (**E, F**) for 2 days. Extracellular HCV titer and intracellular HCV RNA level were determined. **(G-H)** Huh7 cells that had been cultured in glucose (**G**) or galactose (**H**) for 12 hours were infected with HCV at MOI of 2 for 1 day, and then treated with 10 ng/ml of IL-1β. Extracellular HCV titer and intracellular HCV RNA level were determined at 12-, 24-, 48- and 72-hour post-treatment. Data were presented as the mean ± standard deviation (error bars) of triplicates. glc: glucose medium; gal: galactose medium.

## MAPK-p38 phosphorylation mediates glycometabolic regulation of HCV release

It has been shown that alcohol intake can induce production of proinflammatory mitochondrial DNA-enriched extracellular vesicles via p38 MAPK-dependent mechanism [41]. This result raised a possibility that the p38 phosphorylation may play a role in this glycometabolism-regulated HCV release. To test this hypothesis, we first determined the p38 expression level and its phosphorylation form in the glucose- or galactose-cultured Huh7 cells. As shown in Fig 6A, while the p38 protein level was comparable between the two media, its phosphorylation level was significantly reduced in the galactose-cultured cells. Consistently, inhibition of glycolysis with 2DG treatment decreased p38 phosphorylation in the glucose-cultured Huh7 cells (Fig 6B), whereas IL-1β treatment increased the p38 phosphorylation in the galactose-cultured Huh7 cells (Fig 6C).

Next, we examined whether down-regulation of p38 would affect HCV release. HCV-infected Huh7 cells in glucose medium were treated with 10 μM SB203580, a p38 inhibitor [42], and the extracellular and intracellular HCV titers were determined at 72-hour after the treatment. As shown in Fig 6D and 6E, SB203580 treatment significantly decreased extracellular HCV titer and increased intracellular HCV titer, while it had little effect on HCV NS3 and Core protein expression (Fig 6F). Consistently, knockdown of p38 expression also reduced HCV release in the glucose-cultured Huh7 cells (Fig 6G–6I). In contrast, ectopic expression of MKK6EE, a constitutively active form of protein kinase MKK6 that catalyzes the phosphorylation of p38 [43], significantly increased HCV release in the galactose-cultured Huh7 cells (Fig 6J–6L). Taken together, these results demonstrated that the MAPK-p38 phosphorylation plays a critical role in glycometabolic regulation of HCV release.

## HCV cell-to-cell transmission is not affected in the galactose medium-cultured cells

HCV spreads throughout hepatocytes in two different routes, the free-virus infection and cell-to-cell transmission [44,45]. It has been shown that both transmission routes need the essential viral entry receptors [44,46]. Therefore, we sought to investigate whether HCV cell-to-cell transmission is affected in the galactose-cultured cells. To do this, we applied anti-HCV neutralizing antibodies (nAbs) known to block free-virus infection but not cell-to-cell transmission [44,46,47]. As shown in Fig 7A, HCV infectivity in both glucose and galactose culture supernatants can be completely blocked by 3 μg/ml of a previously described HCV E2-specific monoclonal nAb 8D6 [48]. To assess the cell-to-cell HCV transmission, Huh7 cells that had been infected with a previously reported recombinant HCVcc expressing GFP-tagged NS5A [49] (producer) were mixed with naïve Huh7 cells (recipient) in a ratio of 1:1000. The producer and recipient cells were co-cultured in the presence or absence of 3 μg/ml of nAb 8D6 for 72 hours, and the size of infected foci was analyzed by counting the number of GFP-positive cells in each focus (Fig 7B). The nAb treatment dramatically decreased the focus size in glucose-cultured cells but had little effect on the focus size in galactose-cultured cells (Fig 7C), suggesting HCV spreads mainly via free-virus infection in the glucose-cultured cells, while its spread in the galactose-cultured cells is likely dependent upon the cell-to-cell transmission. Importantly, there was no difference in the focus size between the two media when nAb was applied, indicating a comparable efficiency of cell-to-cell HCV transmission between the two media.

## Discussion

HCV cell culture (HCVcc) system mainly uses Huh7 or derivative hepatoma cell lines for efficient virus propagation. Like other tumor cell lines, these cells are normally cultured in

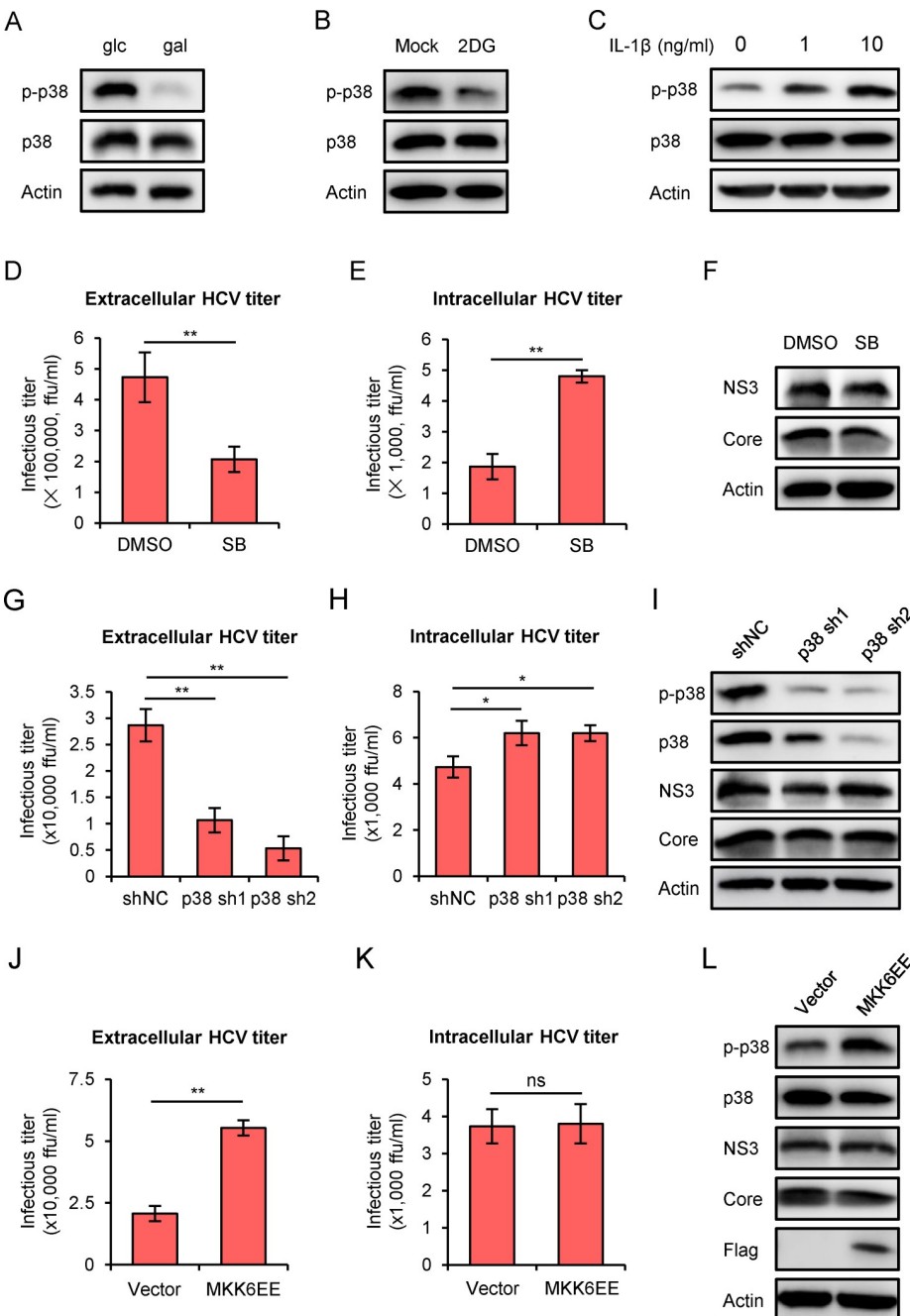

**Fig 6. MAPK-p38 phosphorylation mediates glycometabolic regulation of HCV release.** (**A**) Western blot of p38 and its phosphorylated form in Huh7 cells cultured in glucose or galactose medium for 48 hours. (**B**) Western blot of p38 and its phosphorylated form in Huh7 cells cultured in glucose medium supplemented with 4 mM 2DG for 72 hours. (**C**) Western blot of p38 and its phosphorylated form in Huh7 cells treated with 0, 1, 10 ng/ml of IL-1β in galactose medium for 2 days. (**D-F**) Huh7 cells that had been cultured in glucose medium for 12 hours were infected with HCV at MOI of 2 for 24 hours, and then treated with 10 μM SB203580 or DMSO control. Extracellular HCV titer (**D**), intracellular HCV titer (**E**) and intracellular HCV NS3 and Core proteins (**F**) were determined at 72-hour after SB203580 treatment. (**G-I**) Huh7 cells stably expressing shRNA targeting p38 (sh1 and sh2) or control shRNA (shNC) were cultured in glucose medium for 12 hours, and then infected with HCV at MOI of 2 for 72 hours. Extracellular HCV titer (**G**), intracellular HCV titer (**H**), p38 (phosphorylated and total), HCV NS3 and Core proteins (**I**) were determined. (**J-L**) Huh7 cells cultured in galactose medium were infected with HCV at MOI of 2 for 24 hours, and then transduced with lentiviruses expressing FLAG-MKK6EE or empty vector control for 72 hours. Extracellular HCV titer (**J**), intracellular HCV titer (**K**), p38 (phosphorylated and total), HCV NS3 and Core proteins (**L**) were

determined. Data were presented as the mean ± standard deviation (error bars) of triplicates. glc: glucose medium; gal: galactose medium. SB: SB203580.

glucose-containing medium and generate adenosine triphosphate (ATP) through aerobic glycolysis [17]. Here, we investigated how the HCV infectious cycle can be affected by culturing Huh7 cells in galactose medium, which forces the cells to rely more on mitochondria to generate ATP through oxidative phosphorylation, a metabolic state that is believed to be adopted by normal cells under physiological conditions. Our results demonstrated that when Huh7 cells are cultured in galactose medium, HCV release but not other steps of virus life cycle is impaired, leading to accumulation of infectious virions within the cells. Interestingly, this metabolic alteration does not perturb general lipid metabolism as well as apoE/VLDL secretion,

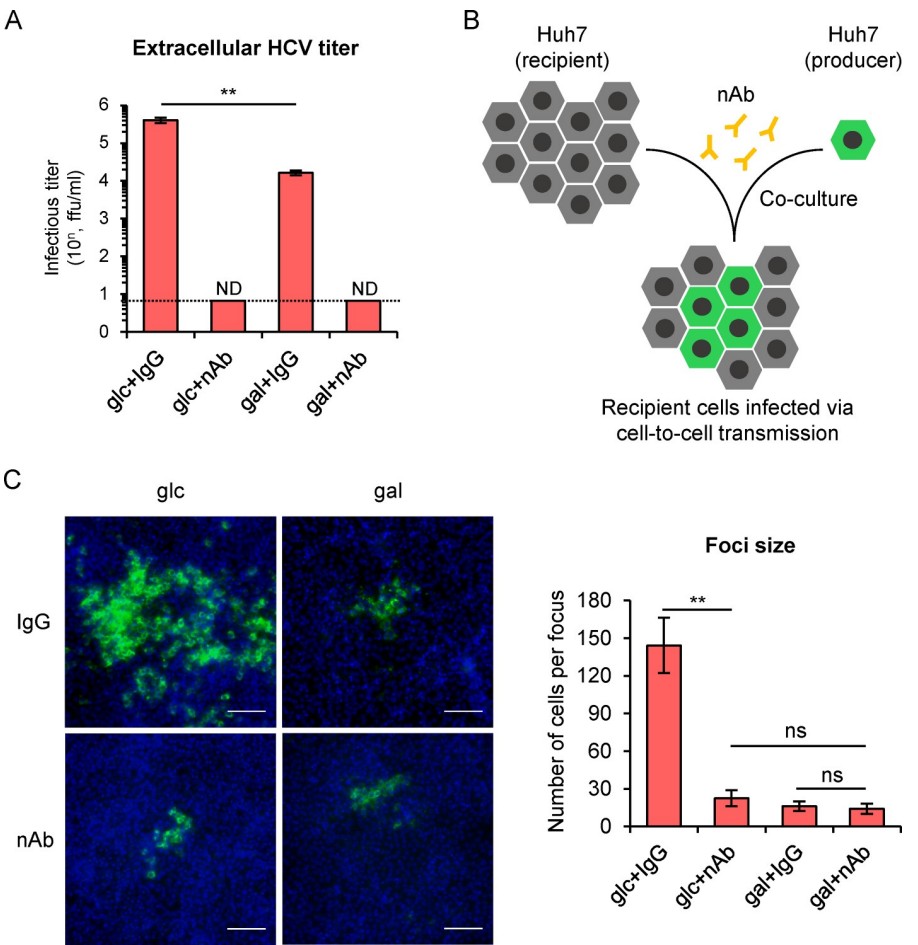

Fig 7. HCV cell-to-cell transmission is not affected in the galactose medium-cultured cells. (A) Huh7 cells cultured in glucose or galactose medium were infected with HCV at MOI of 2 in the presence of anti-E2 neutralizing antibody (nAb, 3 μg/ml) or control IgG. The extracellular titers were examined on day 3 post-infection. ND: not detected; Dashed: detection limit. (B) Schematic of the HCV cell-to-cell transmission assay. Huh7 cells that had been infected by GFP-tagged HCVcc (producer) were mixed at a 1:1000 ratio with naïve Huh7 cells (recipient) at a seeding density of 90% confluency. nAb or control IgG was added into culture medium to neutralize viruses in the culture supernatants. The cell-to-cell virus spread was determined by counting the number of GFP-positive cells in each focus on day 3 post-coculture. (C) A representative fluorescent image of GFP-positive cell focus (bar 100 μm). The average focus size of each treatment group was shown on the right. The error bars represented standard deviation from the measurements in at least 10 foci.

suggesting that HCV release and VLDL release are two independent processes. We also noted that the supernatant HCV specific infectivity was lower in galactose medium. We do not think the decreased infectivity or specific infectivity in galactose medium is caused by the reduced ability of virions to infect the cells, since the ability of intracellular virions to infect the cells is not compromised. On the contrary, the intracellular infectivity increases in the galactose-cultured cells. We speculate that glycometabolism specifically regulates the release of fully assembled infectious virions but may not regulate the release of those non-infectious viral RNA.

In contrast to Huh7 cells, primary human hepatocytes are more likely to use oxidative phosphorylation for energy metabolism. Therefore, it would be interesting to test whether HCV release in these cells can be enhanced by shifting the metabolism towards glycolysis, for example, by treatment with EGF or TNFα, manipulating PKM2 dimeric formation with pTyr or inhibiting oxidative phosphorylation in the primary human hepatocytes [40].

Our results demonstrated that intracellular infectious HCV virions are trapped in MVB of the galactose-cultured cells. This conclusion is supported by several lines of evidence. First, immuno-gold labeled HCV-like particles are found within MVB (Fig 2D). Second, intracellular infectious virions are found to cofractionate with CD63-positive MVB-like vesicles in the galactose-cultured cells (Fig 3). Interestingly, majority of the intracellular infectious virions in the glucose-cultured cells are also found in fraction #8, strongly suggesting that MVB is the reservoir of intracellular fully-assembled HCV virions that are ready to be released regardless of the metabolic state in hepatocytes. The observation that less intracellular HCV virions are found in the glucose-cultured cells is likely due to less MVBs accumulation in the cells. Our results are in line with a previous report that infectious viral particles accumulate in CD63-positive exosomal structures and large dysfunctional lysosomal structures when the exosome release is blocked by a chemical inhibitor U18666A [13]. We speculate that fully-assembled nascent HCV virions enter MVBs through two ways. The first possible way is that the virions are directly sorted into MVBs/late endosomes. This possibility is supported by many reports that the components of endosomal sorting complex required for transport (ESCRT) system are required for HCV release [12]. The second possible way is that the virions may enter early endosome which in turn matures into MVBs/late endosomes. Interestingly, cell culture in galactose medium has no impact on the release of closely-related flaviviruses ZIKV and DENV (S4 Fig), suggesting that there may exist some mechanisms to selectively transport HCV into MVBs. Further investigations are needed to uncover the underlying molecular mechanisms.

MVBs are specialized form of late endosome which can be destined to fuse with lysosomes for degradation or fuse with plasma membrane for exosome release [32]. Lysosomal degradation defects, perturbation of lysosome or prevention of MVB-lysosome fusion have been shown to promote exosome secretion [37,50,51]. Consistently, we found apilimod, an inhibitor of PIKfyve responsible for the lysosome targeting of MVB, enhances HCV release in the galactose-cultured cells but not in the glucose-cultured cells. These data suggest that reduction of HCV release in the galactose-cultured cells is likely due to a general impact of the galactose-induced metabolic state in the cell that tends to shift the outbound trafficking of MVB to the inbound movement toward lysosome. Furthermore, we found that pro-inflammatory cytokines could specifically enhance HCV release in the galactose-cultured cells. The enhanced HCV release might be due to the upregulation of glycolysis stimulated by pro-inflammatory signaling pathways as previously reported [52]. Although TNF-α has been reported to inhibit the spread of HCV among liver cells [53], we did not find the inhibitory effects of TNF-α or IL-1β on HCV release when the glucose medium-cultured Huh7 cells were infected with HCV at a high MOI. Instead, we found that TNF-α and IL-1β dose-dependently enhance HCV release in the galactose medium-cultured Huh7 cells (Fig 5A–5D). More studies will be needed to find out whether these proinflammatory cytokines have any impact on other steps of HCV

life cycle, as we observed a minor inhibition of intracellular HCV RNA levels in the TNF-α-treated cells. Nevertheless, our finding points out the potential regulation of HCV release by inflammation in patients and may provide new insights into the pathogenesis of HCV. It would be of interest to investigate in the future how glycometabolism and inflammation regulate the fate of MVBs in the cell.

We found that the glycometabolism-regulated HCV release is mediated by p38 phosphorylation. Our results showed that p38 phosphorylation and HCV release were reduced in Huh7 cells cultured in galactose medium or treated with glycolysis blocker 2DG, but enhanced by IL-1β treatment (Figs 1, 5 and 6), hinting a correlation between p38 phosphorylation and HCV release. Furthermore, we showed that HCV release can be reduced by a chemical inhibitor or shRNA targeting p38 (Fig 6D–6I), suggesting p38 plays a role in regulating HCV release. Besides, a constitutively active form of MKK6 known to catalyze p38 phosphorylation can promote HCV release in the galactose-cultured cells (Fig 6J–6L). These results suggested that reduced p38 phosphorylation level is the cause of HCV release blockade in the galactose-cultured cells. It has been reported that glucose activates p38 phosphorylation in glomerular mesangial cells, and this effect is mediated by $O$-linked β-$N$-acetylglucosamine ($O$-GlcNAc) produced during glycolysis [54]. This mechanism may explain our observations that p38 phosphorylation is reduced in Huh7 cells cultured in galactose medium or treated with glycolysis blocker 2DG. Ma, J. *et al.* found that p38 inhibition can downregulate expression levels of several genes related to exosome biogenesis and secretion, such as RAB27A, RAB27B, RAB11A and SMPD3 [41]. It would be of interest to investigate which one(s) of these p38 substrates participate in HCV release process. Interestingly, a previous study demonstrated that HCV infection can upregulate p38 phosphorylation level [55]. This suggests that HCV may exploit this mechanism to promote its own secretion.

Cancer cells secrete much more exosomes than normal cells in order to transfer growth signals to remote cells and to suppress immune cell function [56], raising a possibility that HCV release from hepatocytes into circulation *in vivo* may be naturally inefficient. Interestingly, we found that HCV cell-to-cell transmission is not affected by glycometabolism, suggesting that HCV cell-to-supernatants release and cell-to-cell transmission are two mechanistically distinct pathways. We hypothesize that the metabolic state in hepatocytes *in vivo* may favor HCV cell-to-cell transmission over virus release into circulation, therefore allowing HCV to effectively evade neutralizing antibodies and host immune surveillance to establish persistent infection.

## Materials and methods

### Cell culture

Huh7, Huh7.5.1, and HEK293T cells were maintained in high glucose Dulbecco's modified Eagle's medium (DMEM) (GIBCO) supplemented with 10% fetal bovine serum (FBS) (LONSA), 100 U penicillin and 100 μg/ml streptomycin (GIBCO), 10 mM HEPES (GIBCO), 2 mM L-glutamine (GIBCO) and MEM Non-Essential Amino Acids Solution (GIBCO) at 37°C in a humidified atmosphere with 5% $CO_2$. For the cell culture involving galactose, none-glucose DMEM (GIBCO) supplemented with the above-mentioned additives and 25 mM D-(+)-galactose (Sigma) was used.

### Virus preparation and quantification of infectivity titers

The preparation and titration of HCV cell culture (HCVcc) was as described previously [16]. For intracellular HCV titer determination, approximately $1x10^6$ cells were lysed in 50 μl of complete medium, the titers were measured and represented as foci formation units per milliliter (ffu/ml). HCV stocks are prepared by infection of Huh7.5.1 with a JFH-1-derived high

titer virus D183 [57]. The titration of ZIKV and DENV was as described previously [58]. ZIKV strain SZ-WIV01 (GenBank number KU963796) was obtained from the Center for Emerging Infectious Diseases (Wuhan Institute of Virology, Chinese Academy of Sciences, Wuhan, China).

### Iodixanol density gradient centrifugation

Optiprep iodixanol density media was purchased from Sigma-Aldrich (D1556). Cells were washed three times in PBS and re-suspended in 0.8 ml hypotonic buffer (10 mM Tris-HCl, 2 mM $MgCl_2$) with protease inhibitor cocktail (Roche). Cells were lysed by 15 passages through a 26-gauge needle and centrifuged at 1000 g for 10 m to remove cellular debris and nuclei. 0%, 15%, 20%, 25% iodixanol gradients were prepared by mixing optiprep density medium with TNE buffer (10 mM Tris-HCl, 150 mM NaCl, 2 mM EDTA) proportionately. 0.5 ml post-nuclear supernatant was mixed with 0.5 ml 60% optiprep density medium to form a 30% opti-prep gradient and was layered on the bottom of a 5 ml ultracentrifuge tube, 25%, 20%, 15%, 0% iodixanol gradients were layered sequentially. Samples were centrifuged at 150,000 g for 5 hours using HITACHI CP80NX. Fractions of 1–12 are collected from top to bottom with every 400 μl collected into one tube.

### Electron microscopy

Galactose or glucose medium-cultured Huh7 cells with and without HCV infection were fixed with 2% paraformaldehyde/0.2% glutaraldehyde fixation solution. The fixed cells were washed with PBS and scraped off the dishes with cell-scrapers and transferred into eppendorf tubes, and cells were centrifuged for 5 min to form cell pellets. Then re-suspend the cells with 10% gelatin and infiltrating for 10 min at 37˚C, and centrifuge for 5 min and discard the supernatant and solidify the gelatin on ice for 30 min. Then cut the solidified gelatin with cells into small blocks (<1 $mm^3$) and infiltrate into 2.3 M sucrose overnight. Transfer the infiltrated specimen block onto a clean specimen stub and immerse in $LN_2$ until freezing is complete. Then the specimens were thin-sectioned with an Ultra microtome UC7/FC7 at -120˚C with 70 nm thickness. The sections were then transferred onto formvar-coated EM grids. For direct EM analysis, the sections were embedded and stained with 2% methyl cellulose/4% uranyl ace-tate staining solution (with a ratio of 9:1) and imaged on a Tecnai T12 microscope (FEI).

For immuno-labeling analysis, the sections were blocked with 1% bovine serum albumin (BSA) in phosphate-buffered saline (PBS) for 20 min. After blocking, the sections were incu-bated with anti-E2 antibody diluted in blocking buffer for 2 hours and washed 5 times with blocking buffer (2 min each). Then the sections were incubated with secondary antibodies conjugated with 10-nm gold for 1 hour. After washing 5 times with PBS (2 min each) and 3 times with water (1 min each), the sections were embedded and stained with 2% methyl cellu-lose/4% uranyl acetate staining solution (with a ratio of 9:1) and imaged on a Tecnai T12 microscope (FEI).

For EM analysis of the contents of fractionated organelles, 10 μl of fraction #4 and #8 were applied to the glow-discharged EM carbon grids and stained with 0.75% (wt/vol) uranyl for-mate. Negatively stained EM grids were imaged on a Tecnai T12 microscope (FEI) operated at 120 kV with an Eagle (CCD) camera.

### Statistical analysis

Comparisons for two groups were calculated by unpaired two-tailed Student's t tests (Micro-soft Excel). Error bars = mean ± standard deviation. *$p < 0.05$, **$p < 0.01$, ns: non-significant.

## Supporting information

**S1 Text. Supplemental methods.**
(DOC)

**S1 Fig. The effect of culturing in glucose or galactose medium on HCV infectious cycle and cell proliferation.** **(A)** Lactate production assay. Huh7 cells were cultured in glucose or galactose medium, and quantified at 24-, 48-, 72- and 96-hour after cell seeding by using the lactate colorimetric assay kit. **(B)** RT-qPCR analysis of HCV RNA levels in Huh7 cells harboring JFH1 subgenomic replicon (SGR) that had been cultured in glucose or galactose medium for 24, 48 and 72 hours. **(C)** Huh7 cells that had been cultured in glucose or galactose medium for 12 hours were infected with HCV at MOI of 2. The extracellular HCV RNA levels in the culture supernatants were examined at 24-, 48-, 72- and 96-hour post-infection by RT-qPCR. **(D)** Specific infectivity was calculated from Figs 1F and S1C. **(E)** Cell titer assay. Huh7 cells were cultured in glucose or galactose medium, and quantified at 0-, 24-, 48-, 72- and 96-hour after cell seeding by using the CellTiter-Glo cell viability assay kit. Data were presented as the mean ± standard deviation (error bars) of triplicates (**A, B, C, E**). glc: glucose medium; gal: galactose medium.
(TIFF)

**S2 Fig. HCV release is rapidly reduced in the galactose medium-cultured cells.** **(A)** Schematic representation of the medium switch experiment. **(B-E)** Huh7 cell samples were collected at 0-, 2-, 12-, 24-, 36-, 48- and 72-hour post-medium switch. The cell number (**B**), extracellular titer (**C**), intracellular titer (**D**) and viral Core protein (**E**) were determined. Data were presented as the mean ± standard deviation (error bars) of triplicates. glc: glucose medium; gal: galactose medium.
(TIFF)

**S3 Fig. HCV release is reduced in the galactose medium-cultured cells.** **(A)** Schematic of the medium switch experiment. **(B-C)** Kinetics of extracellular and intracellular HCV titers following medium switch. Huh7 cells were infected with HCV at MOI of 2 in glucose medium. The extracellular and intracellular HCV titers were measured at 2-, 12-, 24-, 36-, 48- and 72-hour after the medium switch. Data were presented as the mean ± standard deviation (error bars) of triplicates. glc: glucose medium; gal: galactose medium.
(TIFF)

**S4 Fig. Galactose medium culturing has no effect on ZIKV and DENV virus release.** Huh7 cells cultured in glucose and galactose medium were infected by ZIKV at moi of 0.05 **(A)** or DENV at moi of 0.01 **(B)**. Extracellular virus titer and intracellular ZIKV-NS3 or DENV-E proteins at 24-, 48- and 72-hour post-infection were examined respectively. Data were presented as the mean ± standard deviation (error bars) of triplicates. glc: glucose medium; gal: galactose medium.
(TIFF)

**S5 Fig. The VLDL secretion and general lipid metabolism are not affected in the galactose medium-cultured cells.** **(A)** Huh7 cells that had been cultured in glucose or galactose medium for 12 hours were infected with HCV at MOI of 2. The VLDL levels in the culture supernatants were measured at 24-, 48- and 72-hour post-infection. **(B)** Huh7 cells were cultured in glucose or galactose medium for 24, 48, 72 and 96 hours. Intracellular and extracellular ApoE levels were examined by Western blot. **(C-E)** The mRNA (**C**) and protein (**D**) levels of SREBP1 or SREBP2, as well as their target gene SCD1, ELOVL6 and ACACA mRNA levels (**E**) in Huh7 cells cultured in glucose or galactose medium for 72 hours. **(F)** Total cholesterol levels were

measured in Huh7 cells cultured in glucose or galactose medium for 72 hours. Data were presented as the mean ± standard deviation (error bars) of triplicates. glc: glucose medium; gal: galactose medium.
(TIFF)

**S6 Fig. MVBs are accumulated in the galactose medium-cultured cells. (A)** Representative transmission electron microscopic images of MVB and intraluminal vesicles (ILVs) in the galactose medium-cultured Huh7 cells with or without HCV infection (bar 100 nm). **(B)** Flow cytometry analysis of CD63 expression in the Huh7 cells cultured in glucose or galactose medium. **(C)** Confocal immunofluorescent images of CD63 (green) and F-actin (red) in Huh7 cells that had been cultured in glucose or galactose medium for 48 hours. Nuclei were stained with Hoechst (blue). **(D)** Quantification of CD63 intensity and distance from the nucleus, n (cell number) > = 10. **(E)** Western blot of the exosome-associated or intracellular CD63 in Huh7 cells cultured in glucose or galactose medium for 48 hours. **(F)** Western blot of the exosome-associated or intracellular CD63 in Huh7 cells cultured in glucose medium supplemented with 4 mM 2DG for 72 hours. glc: glucose medium; gal: galactose medium.
(TIFF)

**S7 Fig. IL-1β treatment promotes exosomal CD63 release in the galactose medium-cultured cells.** Western blot of the exosome-associated or intracellular CD63 in Huh7 cells cultured in galactose medium treated with or without IL-1β (10 ng/ml) for 48 hours.
(TIFF)

## Author Contributions

**Conceptualization:** Tao Yu, Qiankun Yang, Jin Zhong.

**Data curation:** Qiankun Yang.

**Formal analysis:** Tao Yu, Qiankun Yang, Yongning He, Jin Zhong.

**Funding acquisition:** Jin Zhong.

**Investigation:** Tao Yu, Qiankun Yang, Fangling Tian, Haishuang Chang, Zhenzheng Hu, Bowen Yu, Lin Han, Yifan Xing.

**Methodology:** Tao Yu, Qiankun Yang, Fangling Tian, Haishuang Chang, Zhenzheng Hu, Bowen Yu, Lin Han, Yifan Xing, Yongning He.

**Project administration:** Jin Zhong.

**Resources:** Jin Zhong.

**Supervision:** Yaming Jiu, Yongning He, Jin Zhong.

**Writing – original draft:** Tao Yu, Qiankun Yang.

**Writing – review & editing:** Tao Yu, Qiankun Yang, Fangling Tian, Yaming Jiu, Yongning He, Jin Zhong.

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
