## [Decision Letter · Decision Letter 0]

17 May 2021

Dear Dr. Zhong,

Thank you very much for submitting your manuscript "Glycometabolism regulates hepatitis C virus release" for consideration at PLOS Pathogens. As with all papers reviewed by the journal, your manuscript was reviewed by members of the editorial board and by several independent reviewers. The reviewers appreciated the attention to an important topic. Based on the reviews, we are likely to accept this manuscript for publication, providing that you modify the manuscript according to the review recommendations. Please note that reviewer #2 comments are attached.

Sincerely,

Glenn Randall

Associate Editor

PLOS Pathogens

Jing-hsiung James Ou

Section Editor

PLOS Pathogens

Kasturi Haldar

Editor-in-Chief

PLOS Pathogens

orcid.org/0000-0001-5065-158X

Michael Malim

Editor-in-Chief

PLOS Pathogens

orcid.org/0000-0002-7699-2064

Reviewer Comments (if any, and for reference):

Reviewer's Responses to Questions

**Part I - Summary**

Reviewer #1: In this paper, the authors report that a change in glyco-metabolism flux regulate the release of HCV particles from a hepatoma cell line. Specifically, when the cells are cultured in galactose as opposed to glucose, viral release but not entry or replication, was significantly inhibited. Under this culture condition or with treatment by a glycolysis inhibitor, the viral particles are trapped inside MVBs and failed to be secreted. Mechanistically, the authors link this phenomenon to the secretion of extracellular vesicles but not the VLDL particles as pathways such as inflammatory cytokine and p38 MAPK pathways that control the former also regulate HCV secretion. Finally, they showed that cell-to-cell spread of HCV particles was not affected by the perturbation of the aerobic glycolysis. Overall this is a strong paper, the observation appears to be novel, the data are supportive of the conclusions and the paper is well written.

Reviewer #2: (No Response)

Reviewer #3: In this manuscript, the authors describe their study of the impact of glycometabolic pathways on the HCV life cycle. Unlike primary cells, tumor cells, like Huh7 and derivatives, use aerobic glycolysis in the presence of glucose, but use (the more primary cell-like) oxidative phosphorylation pathway in the presence of galactose. The authors show that galactose greatly reduced the release of infectious virus, without impairing any other viral life cycle event. The further demonstrate that under this condition, virus is trapped inside intracellular MVBs, whose fusion with lysosomes further decreases HCV titers and can be stimulated to be released from cells by IL-1beta. Finally, they present evidence that galactose only impacts cell-free, but not cell-cell HCV spread.

This is a well-planned and carried out study, and the manuscript is very well written. The topic is important, and the finding are interesting and justified. This reviewer has the following questions/concerns:

**Part II – Major Issues: Key Experiments Required for Acceptance**

Reviewer #1: 1) Given the primary hepatocytes likely use oxidative phosphorylation for energy metabolism, is it possible to manipulate that to perhaps increase infection/release efficiency of HCV in these cells? Are there ways to shift the balance more towards aerobic glycolysis? A way could be to manipulate PKM2 etc?

2) The precise mechanism by which the galactose switch traps particles in the MVB is not understood. The authors speculate that a general shift of outbound versus inbound traffic is responsible, maybe they should examine this shift and demonstrate that it indeed occurs with experiment in this model system.

Reviewer #2: (No Response)

Reviewer #3: This is a well-planned and carried out study, and the manuscript is very well written. The topic is important, and the finding are interesting and justified. This reviewer has the following questions/concerns:

In Suppl Fig S1, can a comparison of glycometabolic pathways between Huh7 cells and primary hepatocytes by included? One could also imagine comparing the impacts of galactose and glucose on infectious virus release from such primary cells, although replication efficiently will limit the utility of this system.

While the impact of galactose on infectious virus release in Fig 1A is striking (>500x at days 4 and 6?), however intracellular RNA differences are only observed at 6 days and is only 4-fold. While the later was still significant, is the results section down playing this too much by merely saying “both significantly lower”. Some perspective on the RT-PCR assay might help. What is the level of detection of this assay? This might say if the day 6 levels were the only ones quantifiable, and thus the lack of a difference prior to this point might not mean anything. Was a standard curve used, which would provide a more reliable means of absolute quantification?

The blot In Fig 1D, actually all such blots involving detection of viral proteins, could use naïve cell controls.

When possible, the specific infectivity of virus should be described. This would be especially helpful to bring together the infectivity and RNA measurements described in Fig 1F and Suppl Fig S1C, respectively.

Quantifying viral proteins produced from subgenomic replicons by immunoblot is an acceptable means to compare RNA replication fitness, but it can be less sensitive to changes than other methods. RNA quantification would be better, but why not use replicons expressing luciferase genes, which are readily available?

This reviewer would prefer a slight variation to the cell-cell spread assay. Here, one could label the recipient cells with a fluorescent marker to be able to differentiate these from producer cells. This would allow one to differentiate between division of producer cells and actual cell-cell transmission.

Just for clarification, what is the glucose/galactose composition of normal DMEM? Would modifying this ratio enhance the amount of virus researchers can generate in their HCV experiments?

**Part III – Minor Issues: Editorial and Data Presentation Modifications**

Reviewer #1: The p38 part appears to be isolated and a bit underdeveloped. How does that connect with the traffic shifting hypothesis? Is the phosphorylation change a cause or result of the shift? This point should be at least addressed in discussion if can’t be easily addressed with experiments.

Reviewer #2: (No Response)

Reviewer #3: (No Response)

PLOS authors have the option to publish the peer review history of their article (what does this mean?). If published, this will include your full peer review and any attached files.

Reviewer #1: No

Reviewer #2: No

Reviewer #3: No

Figure Files:

Data Requirements:

Reproducibility:

References:

---

## [Editor Report · Decision Letter 1]

23 Jun 2021

Dear Dr. Zhong,

We are pleased to inform you that your manuscript 'Glycometabolism regulates hepatitis C virus release' has been provisionally accepted for publication in PLOS Pathogens.

Best regards,

Glenn Randall

Associate Editor

PLOS Pathogens

Jing-hsiung James Ou

Section Editor

PLOS Pathogens

Kasturi Haldar

Editor-in-Chief

PLOS Pathogens

orcid.org/0000-0001-5065-158X

Michael Malim

Editor-in-Chief

PLOS Pathogens

orcid.org/0000-0002-7699-2064
---

## [Editor Report · Acceptance letter]

8 Jul 2021

Dear Dr. Zhong,

We are delighted to inform you that your manuscript, "Glycometabolism regulates hepatitis C virus release," has been formally accepted for publication in PLOS Pathogens.

Best regards,

Kasturi Haldar

Editor-in-Chief

PLOS Pathogens

orcid.org/0000-0001-5065-158X

Michael Malim

Editor-in-Chief

PLOS Pathogens

orcid.org/0000-0002-7699-2064